# Drivers of urban biodiversity in Mexico and joint risks from future urban expansion, climate change, and urban heat island effect

**Julián A. Velasco[1], Carlos Luna-Aranguré[1], Oscar Calderón-Bustamante[1], Alma Mendoza-Ponce[2], Francisco Estrada[1,2,3], Constantino González-Salazar[1,4] ***

1 ICAyCC-Instituto de Ciencias de la Atmósfera y Cambio Climático, Universidad Nacional Autónoma de México, Mexico City, Mexico, 2 PINCC-Programa de Investigación en Cambio Climático, Universidad Nacional Autónoma de México, Mexico City, Mexico, 3 Institute for Environmental Studies, VU Amsterdam, Amsterdam, the Netherlands, 4 C3-Centro de Ciencias de la Complejidad, Universidad Nacional Autónoma de México, Mexico City, Mexico

* cgsalazar@atmosfera.unam.mx

**Data Availability Statement:** Data and R codes are publicly accessible using the following link:

## Abstract

Urbanization is a phenomenon where humans concentrate in high densities and consume more per capita energy than in rural areas, imposing high pressures on biodiversity and eco-system services. Although Mexico is recognized as a megadiverse country and there is an understanding of ecological and evolutionary processes underlying this high diversity, only some efforts have been devoted to understanding how urban biodiversity has been shaped. Here, we compiled a set of socioeconomic and ecological variables to explore macroecological patterns in urban biodiversity across Mexican municipalities. Specifically, we tested the species-area relationships (SAR) between rural and urban areas across municipalities and evaluated the relative role of different socioeconomic and ecological variables driving urban species richness for terrestrial vertebrates. Finally, we explored the exposure of Mexican municipalities to future urban expansion, the urban heat island (UHI) effect, and climate change. Urban and rural settlements show differences in the shape of SAR models. We found that urban area, size of the network of urban protected areas, the number of ecoregions, and GDP explained the urban total species richness relatively well. Mexican cities in the northeast region may be at a higher risk than others. Based on our analyses, policy-makers should identify priority urban conservation sites in cities with high species richness and low urbanization development. These actions would alleviate future urban biodiversity loss in these growing cities.

## Introduction

Urbanization is a social phenomenon where humans concentrate in high densities and consume more energy per capita than in rural areas [1–3]. Urban sprawl generates notable increases in resource demands that are satisfied through a high human impact on adjacent or distant natural resources in rural areas. One expected result of the urbanization processes is

https://figshare.com/articles/dataset/Mexican_urban_biodiversity/14881755.

**Funding:** We have removed the Programa de Investigación en Cambio Climático at UNAM (PINCC-UNAM) acknowledgment from the manuscript. Funding Statement should be updated as: 1. Universidad Nacional Autónoma de México 2. The Programa de Investigación en Cambio Climático at UNAM (PINCC-UNAM), grant number "n/a" The funders had no role in study design, data collection and analysis, decision to publish, or preparation of the manuscript.

**Competing interests:** The authors have declared that no competing interests exist.

concomitant habitat loss, which drives negative consequences on biodiversity, such as population decline and local extinction [4–7]. However, some species are favored by urbanization insomuch by increasing their abundance (e.g., non-native species) [8], whereas others exhibit traits that allow a successful establishment in these new settings [9, 10].

The ecological, evolutionary, and socioeconomic processes driving urban biodiversity patterns are relatively well-known in cities from higher-income countries [6, 7, 11–15]. By contrast, in lower-income countries, mainly located in biodiversity hotspots, there is a lack of understanding of how these processes shape biodiversity across rural-urban gradients [14, 15]. As urban areas in these countries are projected to increase notably in the following decades [16–18], it is necessary to develop a research agenda about the mechanisms underlying urban biodiversity [15, 19]. Eco-evolutionary research on urban biodiversity is critical to design specific conservation actions. Such strategies should help reduce further biodiversity loss and improve the sustainable agendas of emerging Latin American cities.

Mexico is one of the megadiverse countries with a high number of endemism and threatened species [20–22]. The topographical complexity, geological history, and climatic variation favor elevated alpha and beta taxonomic diversity for several taxa, including terrestrial vertebrates, across the country [23, 24]. Although several human stressors impact biodiversity in Mexico at different spatial scales, habitat loss, and climate change are considered the main extinction drivers in the past and the projected future [25–28]. However, the impacts of urbanization on biodiversity across Mexican cities are poorly understood, and how the interplay of ecological and socioeconomic processes plays a role in maintaining the current levels of urban species richness.

Environmental, geographical, and socioeconomic factors directly influence urban species richness across cities in high-income countries [15, 29, 30]. The ongoing urban sprawl, regional climate change, pollution, and urban heat island (UHI) are the most critical threats to urban biodiversity in Latin American cities [14, 31–33]. However, there is a lack of studies evaluating how these threats act synergistically to impact urban biodiversity in Mexican municipalities.

Recent studies have found that size of urban settlements strongly predicts urban development and intensive resource consumption in Mexico [34]. The urban area also plays a significant role in species richness, since larger metropolitan areas will have more extensive green spaces [14]. The area effects on urban species richness derive from theoretical expectations from species-area relationships (SAR) models [35] and the equilibrium theory of island biogeography [36]. For instance, there is a positive relationship between the size of urban areas and bird species richness at local and regional scales in some Mexican municipalities [37–39]. However, whether the same effects are maintained in other taxonomic groups (e.g., amphibians, reptiles, and mammals) is unknown. In addition, the type of SAR models can inform us about the interplay of different ecological and evolutionary processes (e.g., stochastic, immigration-extinction dynamics, and adaptive evolution [35, 36, 40]. Inspecting SAR curves across an urban-rural dichotomy is one of the first steps to establish whether urban assemblages are unique or an extension of those observed in other less transformed landscapes [41, 42]. For instance, Uchida [42] proposed theoretical scaling relationships between biodiversity and city size, which should vary between urban and non-urban settings exhibiting different slopes and intercepts. SAR for the urban environment usually has more alpha diversity than rural areas, and species increase quickly with city size (i.e., high intercepts and slopes). However, few studies are known across multiple municipalities in the same country or region [41, 43]; therefore, the expected pattern in cities, particularly which is the best SAR model describing the observed data, is poorly understood. In addition, recent work has shown that no single best model fits different datasets. Therefore, an approach that tests several models is likely the best strategy for analyzing this information [44–46].

Here, we compiled an extensive dataset of urban species richness for terrestrial vertebrates and socio-economic and ecological information for 389 municipalities across Mexico to test potential drivers of urban biodiversity. First, we evaluate whether there are differences in species richness across urban and rural settlements using different SAR models. Then, we implemented a machine learning model to assess the relative importance of various socio-economic and ecological drivers explaining urban biodiversity (Table 1). Finally, we explored the future exposure of Mexican municipalities to projected urban expansion, the urban heat island, and climate change. To our knowledge, this is the first study testing how socioeconomic and ecological factors explain urban species richness of terrestrial vertebrates across an extensive dataset for Mexican cities.

## Methods

### Datasets

We obtained occurrence records for four terrestrial vertebrate groups (amphibians, reptiles, mammals, and birds) for Mexican municipalities from the SNIB (Sistema Nacional de Información sobre Biodiversidad) repository [47], which contains information for 2,731 species with 6,763,073 records (Fig A in S1 Appendix) (available in https://www.snib.mx). A great effort was made to sample these taxa across the country, and intensive curatorial and taxonomic work has been carried out for several decades [48–50]. However, this database has many taxonomic and georeferencing errors. Consequently, we eliminated duplicated, erroneous, and doubtful records to maintain those with approximately a spatial precision of approx. 1 km [51]. Furthermore, we harmonized the taxonomic nomenclature from this database with updated international taxonomic treatments for amphibians and mammals from the [52], reptiles from [53] and birds from [54]. Although the sampling effort is unbalanced across the four taxa, birds emerged as more densely sampled/observed (Fig B in S1 Appendix).

We used the urban localities shapefile from the Instituto Nacional de Estadística y Geografía (INEGI), which are denominated as área geoestadística básica (AGEB) from the national demographic census in 2020 (available in https://www.inegi.org.mx/default.html). Urban localities are defined based on human population counts larger than 2,500 inhabitants and the municipal capital, regardless of the inhabitants' number (Fig C in S1 Appendix). We calculated the urban areas for each municipality in Mexico as the sum of the area of their AGEB urban polygons. Only 2287 out of 2463 municipalities contain urban polygons. Municipality is the second-level administrative division in Mexico and differs from cities or towns because they are distinct entities and do not share geographical boundaries. Rural areas were defined as those that have not been urbanized. Under this definition, we included multiple human-modified land uses in the rural category. We performed a spatial join between species' occurrences and urban and rural polygons using the *point.in.poly* function from the spatialEco R package v 2.0.1, and we calculated the number of species for each higher taxon for urban and rural areas (Table A in S1 Data). We used the municipality as a unit of analysis due to the availability of socioeconomic data at this level, which was compiled from Haro et al. 2021 [55]. Haro et al. (2021) compiled data from several databases from Mexican institutions including Sistema Nacional de Información Municipal (http://www.snim.rami.gob.mx), Consejo Nacional de Población (https://www.gob.mx/conapo), Secretaría de Agricultura y Desarrollo Rural (https://www.suri.agricultura.gob.mx:8017/buscadorBeneficiario; https://www.gob.mx/siap), and CONABIO (http://www.conabio.gob.mx/informacion/gis/) (further details in [55]). Additional environmental data were collected for each municipality's urban areas from additional multiple sources available publicly [27, 56–59] (see Table 1).

**Table 1. Ecological and socioeconomic processes approximated by a series of variables that are linked with urban species richness in Mexican cities.**

| Hypothesis | Variable | Prediction | Explanation |
|---|---|---|---|
| **City size** | Urban area (log10Area) | Larger cities are expected to maintain more species than smaller cities | Species-area relationships (SAR) can be interpreted under MacArthur & Wilson's [36] equilibrium theory of island biogeography theory. Larger cities are expected to exhibit more species, but some thresholds can emerge in SARs [35]. These thresholds are likely related to a minimum city size requirement to maintain multiple species. |
|  | Gross domestic product (GDP) | Cities with higher GDP are expected to maintain more species than cities with lower GDP. | A proxy of economic development that might be related to government and private investment in more green spaces and economic resource allocation to implementation of specific urban conservation strategies. |
|  | Human total density (Dens_total) / Human rural density (Dens_urban) | We expect municipalities with high human total and rural density to contain a smaller number of species. | Species richness and human density are positively correlated at coarse-grain spatial scales but negatively at fine-grain scales [87, 88]. The positive relationship at larger scales is mediated by energy availability because human settlements are in high-energy regions where resource availability is larger [89]. |
| **Habitat Intactness** | Urban tree height (TreeHeight) | Cities with higher tree heights are expected to maintain more species than cities with lower tree heights. | Tree height is a proxy of available green space, either as urban protected areas or urban parks in each city. |
|  | Natural protected areas (area) (log10_ANPs_areas) | Cities with higher protected areas are expected to maintain more species than cities with smaller protected areas. | The cumulative area for the Protected Area Network system in each city is a proxy of high species richness and better levels of conservation management. |
|  | Remnant vegetation (VegtRem) | Municipalities with higher remnant vegetation are expected to have adjacent urban areas with more species. | Remnant vegetation is a strong driver of species richness [90]. |
| **Human impact** | Marginalization | Municipalities with higher marginalization are expected to exhibit fewer species richness. | The marginalization index integrates different elements of poverty, education, housing, population distribution, and incomes in rural areas [55]. Higher marginalization means more social vulnerability in rural areas, and it is expected that more marginalized municipalities overlap with more biodiversity Mexican hotspots [91–93] |
|  | Percentage of the population working in the primary sector (porPob) | Municipalities with more people in the sector are expected to have urban areas with higher species richness. | This metric refers to the percentage of people relying on agricultural production for self-consumption and income. This is a proxy of more rural areas in each municipality and can be linked with high species richness in urban areas by mass effects [94], rescue effects [95], or source-sink metacommunity dynamics [96, 97] |
|  | Demographic pressure index (IPD) | We expect that municipalities with higher demographic pressure are expected to contain fewer species. | More people in rural-urban areas are exerting more pressures on natural resources and modifying land use, which promotes a less accumulation of species in rural-urban areas. |
| **Habitat diversity** | Topographic complexity (TPI) | Urban cities located in more topographic complexity regions are expected to exhibit higher species richness. | Mountain systems with higher topographic complexity tend to exhibit many species [98, 99]. |
|  | Habitat uniformity (Evenness) | Cities with greater habitat heterogeneity are expected to have more species. | Habitat heterogeneity is recognized as a direct driver of species richness across space and time. Habitat variation facilitates more niche space and, therefore, increases the number of co-occurring species in a site [100–102]. |

(*Continued*)

**Table 1.** (Continued)

| Hypothesis | Variable | Prediction | Explanation |
|---|---|---|---|
| | Number of ecoregions (NumbEcoreg) | Cities that are located in regions that overlap different ecoregions are expected to have more diverse biotas and, therefore, more species. | Ecoregions are units containing a distinct species assemblage and environmental conditions [103]. |
| Climate | Annual mean temperature (bio1), Maximum temperature of warmest month (bio5), Minimum temperature of coldest month (bio6), total precipitation (bio12), Precipitation seasonality (bio15). | We expect that municipalities with higher temperature, more rainfall, and low seasonality are expected to have more species | Several potential mechanisms are underlying the climate-species relationships [89, 103]. For instance, regions with higher productivity (i.e., warm and humid climates) and lower seasonality (stable climates around the year) concentrate more species than other regions [104, 105]. |
| | Cloud cover | Cities with a higher percentage of cloud cover are expected to have high species richness. | Cloud cover is a direct proxy for rainfall in mountain regions. Itis linked with important demographic processes on several taxa and strongly predict species richness in tropical areas. Mean annual cloud frequency (%) was calculated for a period over 2000–2014. |

We conducted a rarefaction analysis to select those municipalities with adequate survey sampling for each taxonomic group. In this way, we avoid the potential effects of sampling bias, which might affect the posterior analysis and interpretations. First, we evaluate the inventory completeness fitting a smoothed species accumulation curve [60] with the method exact of the function *specaccum* in the R package vegan [61]. We used the slope of this curve as an index of how many new species appear in urban and rural areas for each municipality and a cut-off value of 0.05 (a flat curve) to exclude those municipalities with incomplete faunistic inventories from the SNIB dataset [60]. These analyses give us a better picture of urban inventories using these opportunistic occurrence datasets from museums and human observations. The dataset of municipalities with adequate biodiversity sampling for urban areas was a composite of 389 Mexican municipalities. We calculated the total species richness (i.e., the sum of species richness for all four taxa) and added the socioeconomic and environmental data previously collected for each of the 389 municipalities (Table A and B in S1 Data). Although several municipalities contain a very low number of species (i.e., less than five species), these vary for each taxonomic group (e.g., for amphibians, there are 66 municipalities with less than five species, but for birds, there are only eight municipalities; see Table C in S1 Data). However, we decided to exclude these municipalities from the species-area relationships and random forest analyses to avoid potential effects on the recovered SAR patterns and the effect size of socio-ecological drivers of urban species richness.

## SAR models

We tested for differences in species-area relationships (SAR) across an urban-rural dichotomy for Mexican municipalities. We log-transformed species richness and area to compare slopes and intercepts between urban and rural environments in the four taxonomic groups. Then, we fit a series of SAR models for urban species richness without transforming data by adopting an information-theoretic framework to compare and evaluate the best functions that maximize the explanatory power of SAR using the small-sampled corrected Akaike information criterion (AICc). We fitted the 20 SAR models in the sars package using the *sar_multi* function [46]. Analyses were conducted for each taxonomic group. The selected models maximized the explanatory power using the Akaike weights derived from the AICc values. We selected the first five models with the highest Akaike weights and evaluated them using the Lilliefors

extension of the Kolmogorov normality test and a correlation test for the residuals obtained from the model-fitted values. We built a multimodel-averaged SAR curve by multiplying the predicted species richness of each model by its Akaike weight and then summing the resulting values [62].

## Random forest models

We used a supervised machine learning Random Forest (RF) algorithm to assess how environmental and socioeconomic variables drive urban biodiversity across the Mexican municipalities (Table 1). RF is a machine learning technique that allows us to evaluate different variables simultaneously by maximizing its predictive prediction power [63]. Furthermore, RF allows us to treat complex multidimensional datasets with complex structures. We trained RF models using total species richness for the four taxonomic groups and 19 variables: total urban area (i.e., log10 urban area), the total size of urban protected areas (log10), number of ecoregions overlapping urban areas, climate (bio1; bio5; bio6; bio12; bio15; cloud cover), topographic complexity, urban tree height, habitat heterogeneity, remnant vegetation, human density, demographic pressure index, and socioeconomic factors (GDP; marginalization; percentage of the population working in the primary sector). A full description of these variables and the hypothesized causality relationships with species richness is in Table 1. We explored various parameters to select the best configuration that minimizes the prediction error using the root-mean-square error (RMSE). A cross-validation strategy was implemented, performing 100 random partitions of each dataset into a calibration (80% of the data) and validation (20%) dataset to avoid RF overfitting and increase model accuracy. Similarly, we quantified variable effects (i.e., effect sizes) for each variable by separating using partial dependence averaging over the effects of all variables with a non-parametric bootstrapping approach [64]. We implemented this approach using the rfUtilities R package [65]. This strategy allows us to interpret more directly how each predictor affects the urban species richness. We analyzed the same effect sizes but controlled urban areas' effects on species richness by using residuals from a log-log linear regression between species richness and urban area.

## Multivariate future risk index

We estimated the exposure to future urbanization across Mexican municipalities using land-use/cover-change (LUCC) scenarios recently developed for Mexico (see [27] for further details). The number of accumulated urban pixels projected toward 2070 within each municipality was calculated using the assumptions of two scenarios, business-as-usual (BAU) and worst scenario (WRT) (Fig D in S1 Appendix). The annual mean temperatures for urban areas in each municipality were calculated using the climatological database from the period 1960–1990 from the Worldclim repository [56]. Future temperature projections for 2070 were obtained from an ensemble of 11 General Circulation Models and two Representative Concentration Pathways (RCP; -RCP2.6 and RCP8.5-) from the CMIP5 database available in the Worldclim repository. The RCP2.6 is an emission scenario that represents the goals of the Paris Agreement and, therefore, limits the increase in global mean temperature to 2˚C [66]. In contrast, the RCP8.5 is a high-emission scenario with a combination of high population and limited technological change to improve energy consumption [67]. We calculated the temperature anomalies between present (1960–1991) and future (2070) for both RCP emission scenarios. This anomaly represents the projected shift in urban temperatures for 2070 in each municipality. Similarly, we calculate the difference between the historical urban scenario from INEGI for 2011 and the future projections for business-as-usual (BAU) and

worst (WRT) scenarios [27]. This difference represents the projected shift in urbanization in each municipality.

We estimated the urban heat island effect (UHI) following [68]. Karl et al. [68] developed a methodology to detect the effect of heat island intensity based on the population size of cities in the United States. Specifically, they derived a regression equation where the urban-to-rural temperature differences were estimated using the urban population. We used the same equation but with population data for Mexican cities from a recent demography census (2020) from the Instituto Nacional de Estadística y Geografía INEGI (https://www.inegi.org.mx). The UHI is a simple approximation based on empirical relationships of the form:

$$UHI = a * UrbanTotalpopulation^b$$

The two fixed parameters, a and b, were calibrated for annual temperatures for US cities with a population exceeding the 100,000 inhabitants (see Table 5 in [68]). We used $b = 0.45$ and a central estimate of $a = 1.74x10^{-3}$. Although many studies quantify UHI based on urban-to-rural temperature differences [69, 70], this is difficult to implement in many countries where meteorological stations are scarce and have a lot of missing data. However, empirical methods, as we use here, were developed to estimate the effect of urbanization on annual mean temperatures, removing the effects of the increasing population (i.e., urbanization) across large scales [68, 71, 72].

We standardized the projected shifts in temperature, urbanization, and current UHI between 0 and 1 and summed these in a single metric of multivariate risk. This multivariate risk metric allows us to highlight which municipalities will be more exposed to a combination of impacts from urban expansion, climate change, and UHI in the future (Table D in S1 Data). We evaluate how regional clusters of Mexican cities are related to this multivariate risk metric and discuss how this can potentially affect urban biodiversity. However, we caution here that increases in urban populations are likely related to additional risks beyond UHI, including poor air quality, noise pollution, increase in artificial lights, waste disposal problems, and reduction in green urban spaces.

## Results

### SAR models

We found strong linearity between species richness and urban area across municipalities with log-log transformation (Fig 1). In addition, we found notable differences in SAR shapes between urban and rural environments, with different intercepts but relatively similar slopes (Fig 1). We found that the slope for amphibians was similar between rural and urban areas, and intercept larger in rural areas. However, amphibians are a poorly sampled group in Mexico and likely are underrepresented in urban settings. The SAR model fitting approach revealed that urban species richness and urban area exhibit a non-linear relationship across municipalities, and convex-up, linear, convex-down, or sigmoidal SAR shapes were recovered across the four taxa (Table 2).

### Random forest models

The RF models showed a relatively good predictive capacity although a relatively low explanatory power for urban species richness of terrestrial vertebrates (Fig E in S1 Appendix). The estimates of the effect sizes revealed that urban area, size of the network of urban protected areas, the number of ecoregions, and GDP were the most important positive determinants of urban total species richness (Fig 2a). By contrast, the percentage of people working in the

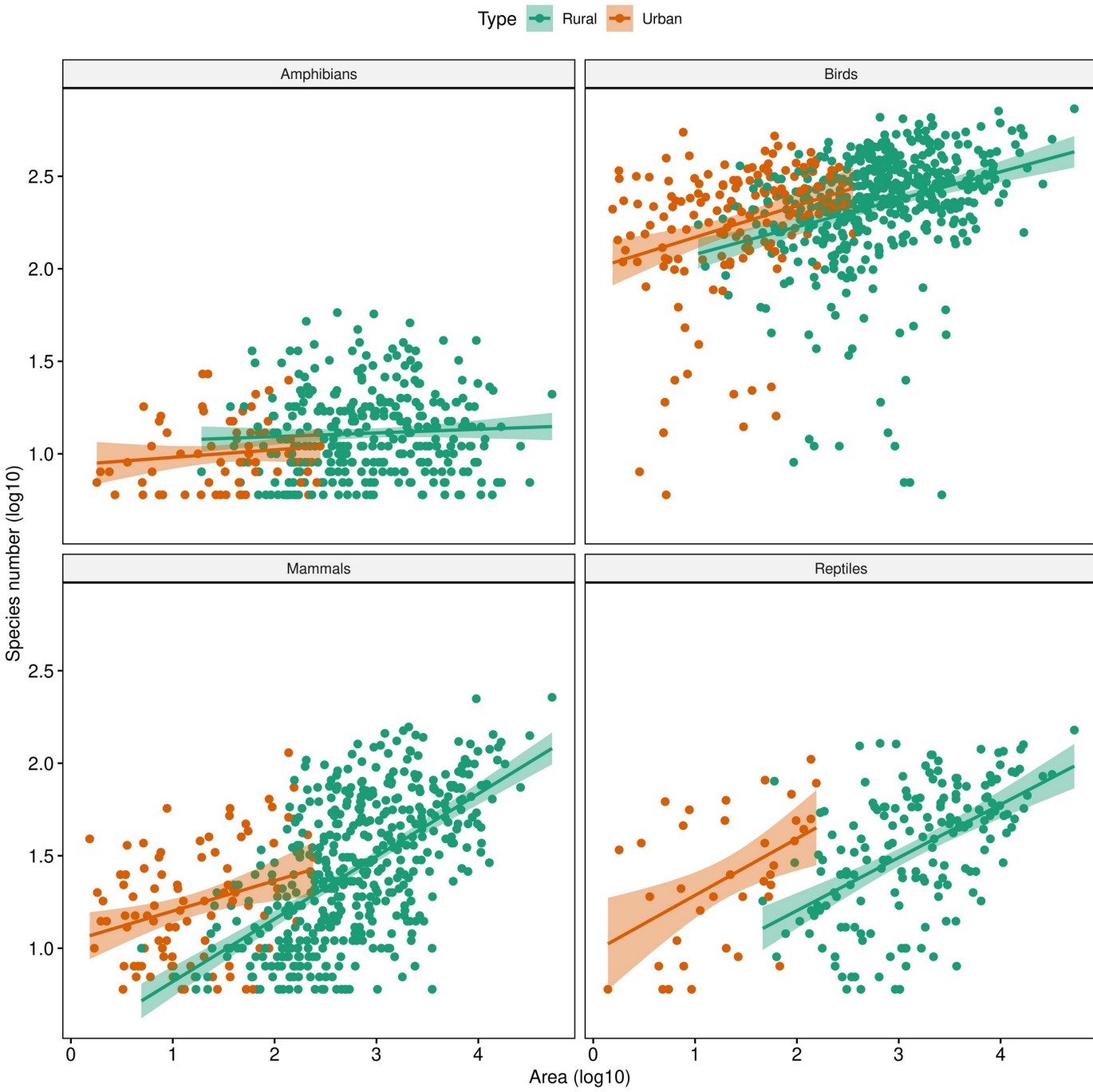

**Fig 1. Species-area relationships (SAR) for terrestrial vertebrates urban/rural assemblages across Mexican municipalities (excluding those with less than five species).** Fitted lines correspond to a log-log transformation model (i.e., power model; see Triantis et al. [44]).

primary sector negatively affects urban species richness across municipalities (Fig 2a). After removing urban area effect on urban species richness, we found a significant effect from topographic complexity but not for number of ecoregions nor the percentage of people working in the primary sector (Fig 2b).

**Table 2. Parameters from species-area relationship (SAR) model explaining urban species richness for four taxonomic groups across Mexican municipalities (excluding those with less than five species).**

| Group | Model | Akaike weights | AIC | R2 | Shape | Asymptote |
|---|---|---|---|---|---|---|
| Amphibians | **Monod** | **0.118** | **-30.821** | **0.038** | **convex up** | **True** |
| | **Negative Exponential** | **0.104** | **-30.580** | **0.034** | **convex up** | **True** |
| | Logarithmic | 0.089 | -30.264 | 0.029 | convex up | False |
| | Koba | 0.089 | -30.264 | 0.029 | convex up | False |
| | Power | 0.087 | -30.206 | 0.028 | convex up | False |
| Reptiles | **Linear** | **0.158** | **27.023** | **0.230** | **linear** | **False** |
| | **PowerR** | **0.156** | **27.051** | **0.280** | **convex down** | **False** |
| | **EPM-1** | **0.153** | **27.084** | **0.280** | **sigmoid** | **False** |
| | Asymptotic | 0.104 | 27.853 | 0.265 | convex down | True |
| | Power | 0.060 | 28.949 | 0.189 | convex up | False |
| Mammals | **EPM-1** | **0.271** | **37.154** | **0.137** | **sigmoid** | **False** |
| | **PowerR** | **0.242** | **37.380** | **0.135** | **convex down** | **False** |
| | Persistence Function 2 | 0.162 | 38.186 | 0.127 | sigmoid | False |
| | EPM-2 | 0.122 | 38.758 | 0.121 | sigmoid | False |
| | Linear | 0.066 | 39.990 | 0.088 | linear | False |
| Birds | **Persistence Function 2** | **0.341** | **107.003** | **0.110** | **sigmoid** | **False** |
| | **EPM-2** | **0.326** | **107.093** | **0.110** | **sigmoid** | **False** |
| | Linear | 0.077 | 109.986 | 0.083 | linear | False |
| | EPM-1 | 0.057 | 110.593 | 0.090 | convex down | False |
| | PowerR | 0.046 | 111.026 | 0.088 | convex down | False |

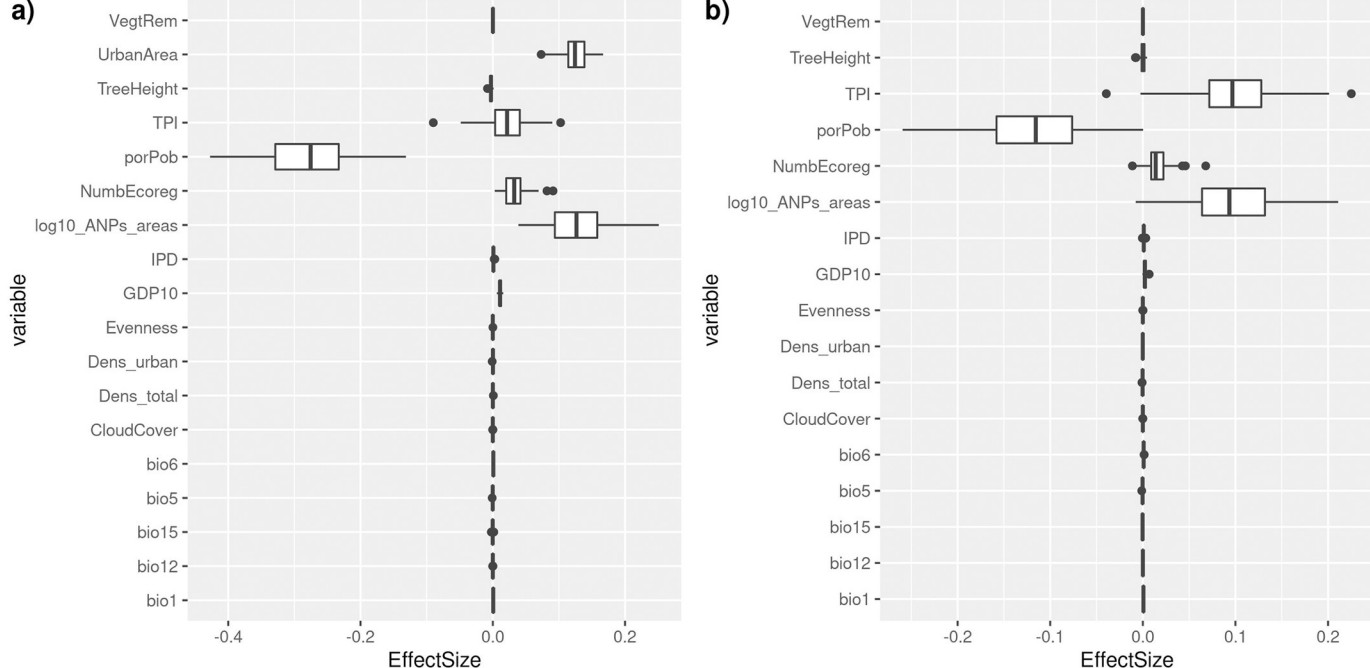

**Fig 2. Effect sizes of socioeconomic and environmental variables driving urban species richness across Mexican municipalities (excluding those with less than five species).** Effect sizes were estimated by using Random Forest models with a cross-validation strategy with 100 random partitions. Panel a) includes urban area and Panel b) includes species richness but controlling for urban area effects. See main text and Table 1 for a full description of variables.

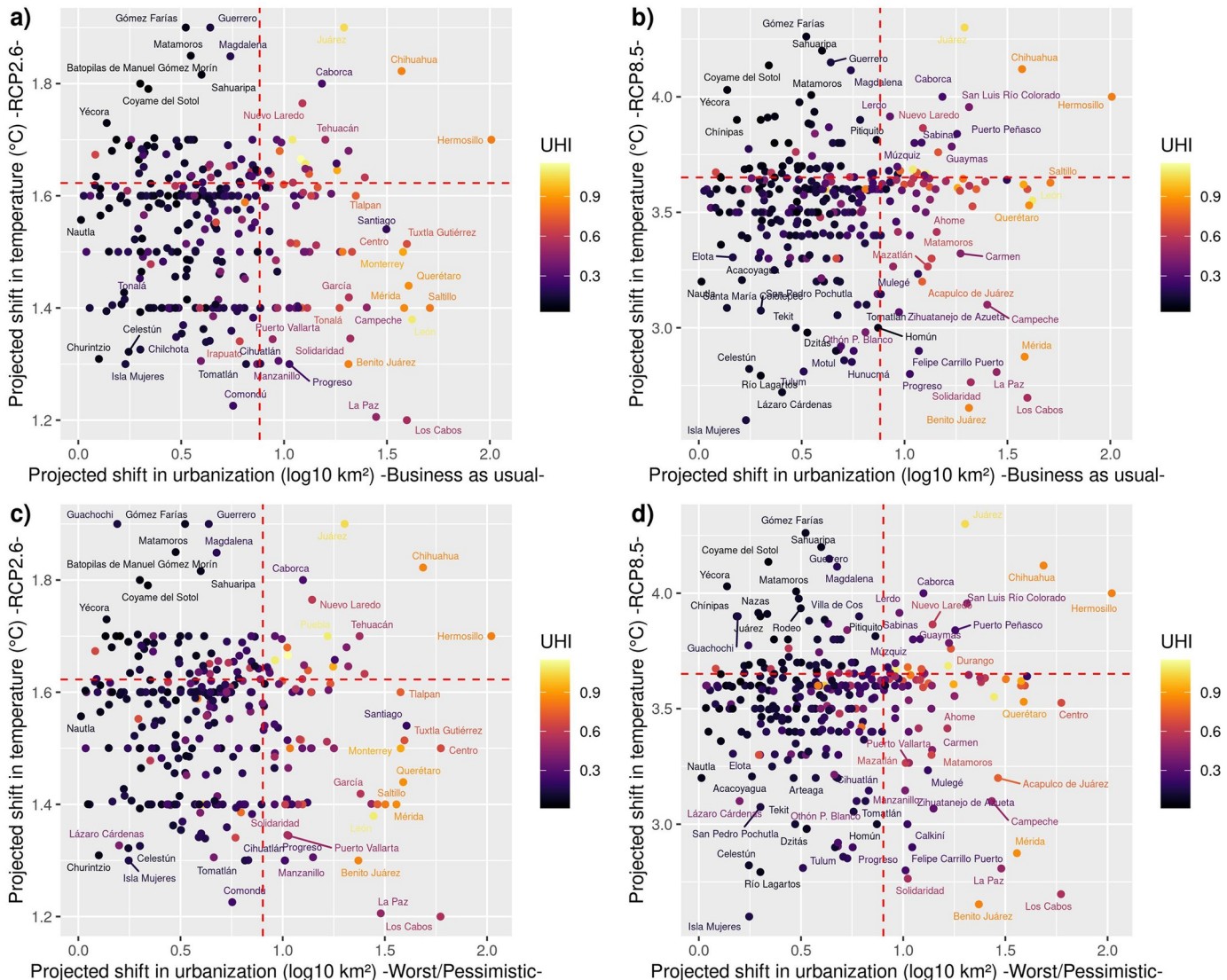

**Fig 3. Projected shifts in temperature, urbanization, and current urban heat island (UHI) effects across 389 Mexican municipalities for two emission scenarios (RCP 2.6 and RCP 8.5) and two land-use change scenarios (Business as usual and worst/pessimistic).** The urban heat island effect was estimated based on the total population (see main text for details). The red dashed lines represent the third quartile value for shifts in temperature and urbanization.

## Multivariate future risk

Municipalities in the north of Mexico (e.g., Hermosillo, Juárez, and Chihuahua) are projected to be the most affected by the combination of increasing temperature, urban expansion, and UHI (Fig 3). Municipalities located in the Valley of Mexico (e.g., Iztapalapa, Puebla, Ecatepec, Gustavo A. Madero) also will face higher risks in the future (Table D in S1 Data). The highest rates of future urban growth occur in municipalities located in different geographic regions (Fig 3). For instance, the worst or pessimistic scenario of urban expansion is projected to occur in Hermosillo, Centro, Los Cabos, Chihuaha, Santiago, Atizapán, Tuxtla Gutierrez, Querétaro, Monterrey, Tlalpan, Mérida, Saltillo, La Paz. Some of these municipalities are located in touristic places (e.g., Los Cabos, Mérida), and it will be necessary to implement urban planning policies to reduce the potential impacts from UHI and climate change. We did not find

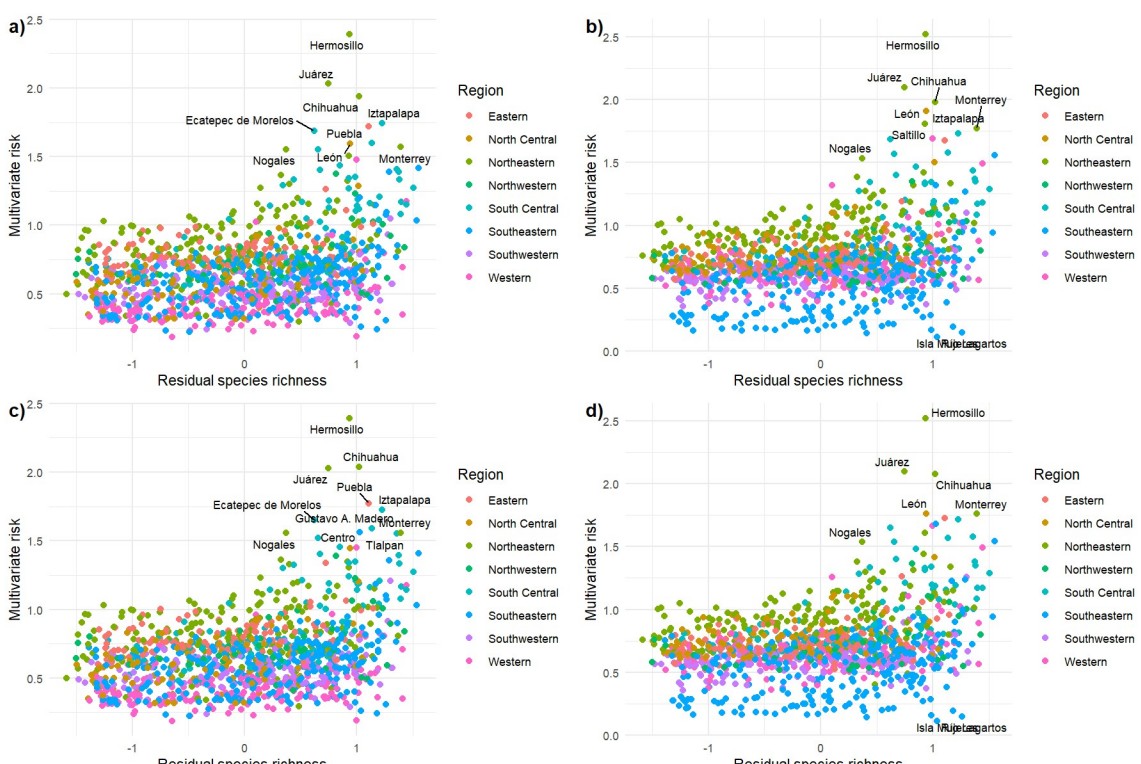

**Fig 4. Scatter-plot of multivariate risks and residual species richness across 389 Mexican municipalities grouped in eight clusters under two land-use change scenarios (business-as-usual and worst/pessimistic) and two greenhouse gases emission scenarios (RCP 2.6 and 8.5) for a time horizon centered in 2070: a) BAU and RCP 2.6; b) BAU and RCP 8.5; c) WRT and RCP 2.6; d) WRT and RCP 8.5.**

substantial differences between different multivariate risk indices from different scenarios (Fig F in S1 Appendix).

The multivariate risk tends to be higher in municipalities in northern Mexico (e.g., Hermosillo, Juárez, and Chihuahua and the central region in Mexico (e.g., Iztapalapa, Puebla, Ecatepec; León; Fig 4). Hermosillo, Juárez, and Chihuahua (located in the northern region of Mexico) exhibited the highest multivariate risk than the other Mexican municipalities (Fig 4), which is driven mainly for future urbanization and UHI than projected shifts in annual mean temperature because there are no substantial differences between the RCP2.6 and RCP8.5 scenarios (Fig 4). The municipalities in the central region of Mexico (i.e., Iztapalapa, Puebla, Ecatepec; León; Fig 4) exhibit high multivariate risk due to projected temperatures and UHI, but not large increments in future urbanization because they are already highly urbanized (Figs 3 and 4; Table D in S1 Data). It is crucial to notice the similarities between the BAU and the WRT scenarios (Fig F in S1 Appendix), which suggest that UHI and future climate change will create increasing and differentiated pressures on biodiversity in large cities (Table D in S1 Data). For instance, the diversification between scenarios in the municipalities located southeast of Mexico is more related to differences between the RCP2.6 and 8.5 than the other two elements. These municipalities, with the highest multivariate risk, have high species richness, largest urban areas, increased protected areas, and large urban density and development (Figs 3 and 4). These municipalities are concentrated in three states (Distrito Federal, Mexico, and Jalisco). Furthermore, we did not find evidence for a geographical pattern in the relationship

between multivariate risk and residual species richness (Figs 3 and 4). After controlling by urban area effects, most municipalities with many species are in different parts of the country (Fig 4). Still, these municipalities are in regions where tropical evergreen and semideciduous forests are dominant, suggesting that the regional species pools likely play a strong role in assembling urban biodiversity through mass effects.

## Discussion

How humans influence the community assembly processes in Latin American cities is a relevant question in urban ecology and conservation today [14, 40, 73]. We found differences in urban and rural assemblages in the species-area relationships across four taxonomic groups. Urban species assemblages of 389 Mexican municipalities are strongly related to scaling effects associated with the size of urban areas, the number of ecoregions present, the network of urban protected areas, human impact degree, and the economy size. Other environmental and socioeconomic variables seem not to play a substantial role in shaping the urban assemblages. Many municipalities with small urban areas contain few species, and likely, these areas are more susceptible to stochastic events than larger urban areas where habitat filtering mechanisms may be more dominant [5, 30, 74].

### Drivers of urban biodiversity

The fitting model approach of SAR models revealed that urban species richness in Mexican municipalities may be described by a sigmoidal curve [75, 76] and several ecological processes may be inferred for small, intermediate, and large urban areas (Fig 1). For instance, municipalities with small urban areas have fewer species than those with larger urban areas regardless of the surrounding habitat, biogeographical region, or specific urban planning policy. Mass effects may play a major role in shaping urban biotic assemblages in tropical regions than those in desert or xeric regions. The municipalities with few species are expected to be influenced by colonization-extinction dynamics, and probably urban assembly is dominated by stochastic events. Accordingly, more ecologically relevant habitat is necessary to allocate in these municipalities to reduce local extinction rates.

Our results are in line with other large-scale studies in Europe, with rural and regional assemblages exhibiting a similar positive relationship between the urban area and species richness (e.g., [41]). A sigmoidal shape in species-area relationships across the four taxonomic groups suggests that differential ecological and evolutionary processes shape larger and smaller areas. For instance, a potential limit to the number of species in larger urban areas (i.e., a potential carrying capacity) suggests that some municipalities' urban biodiversity is likely already homogenized. Therefore, it is expected that biotic composition is more similar in these places. Further studies will be necessary to evaluate whether biotic homogenization emerges across Mexican municipalities and how urban configurations shape urban ecological patterns.

The Random Forest (RF) models corroborated the results of the SAR model fitting (Fig 2). RF is a highly flexible method that allows comparing the relative role of different drivers of urban species richness (see [55]) for a similar approach by using maize yields). We found evidence of scaling effects on urban species richness through different variables related to the urban area's size. For instance, GDP is a proxy of urban development and urban area [34]. Therefore, it is expected that GDP will be a proxy of more available economic resources from private and public institutions to invest in creating urban green spaces, translating into a high number of species (Table 1). However, municipalities with high GDP are not necessarily reflecting a high and homogeneous income in the population. Accordingly, more research is necessary to establish how income differences inside urban areas across municipalities impact

biodiversity at finer scales in large urban conglomerates (e.g., in the central Valley of Mexico). Although GDP showed a null effect in RF models after control by area in species richness, fine-scale wealth and socioeconomic data probably have more explanatory power than GDP at the municipality level. There is a well-established relationship between local biodiversity and economic wealth for multiple taxa [77–79], and therefore, a similar pattern may be discovered in future studies with more appropriate economic and wealth variables. Similarly, the negative effect of the percentage of people working in the primary sector (porPob; Fig 2) suggests that municipalities with high social marginalization likely have fewer green spaces and fewer species.

The lack of importance of other variables is likely due to the coarse-grain scale of the available data (Table 1). We believe that finer-grain data, including microclimatic variation, size of urban green spaces (e.g., small-intermediate green spaces with less than 1 ha), landscape connectivity, local vegetation diversity, and socioeconomic and wealth indicators at the AGEB scale, help us to understand better what drives urban biodiversity in Mexican municipalities. In addition, data about the historical trajectories of Mexican cities (e.g., past urban dynamics and ages) would allow us to evaluate whether urban biodiversity followed similar courses as in Europe [7].

Further research on local biotic inventories is necessary to understand the local processes driving urban species richness. The interplay of different scale-dependent processes shapes urban ecological patterns, and more work on this is required to answer questions oriented to establish how species respond to local urban environments. Further extensive city-scale fieldwork, ideally across urban-rural gradients, could help elucidate how current urban biodiversity can be maintained or improved and their emergent ecosystem services upon specific urban green space configurations.

## Multivariate risk index

Our results show that increasing temperatures produced by the urban heat island and urban expansion could impact urban biodiversity and exacerbate climate change impacts at local scales. Municipalities in northern Mexico should be considered as a priority for local mitigation and habitat recovery due to high exposure to future climate change, urban sprawl, and the urban heat island. These municipalities have faced stronger temperature increases due to contemporary climate change than other regions [80]. However, we consider that it is necessary to conduct a comprehensive assessment of how urbanization affects local biodiversity across spatial scales in these cities, which are projected to grow quickly in the next decades. National studies, as we conduct here, allow us to identify the main trends of land-use change and critical resource areas; however, more detailed local-scale studies are needed for effective monitoring programs and developing local conservation strategies to avoid further biodiversity loss. For instance, the identification of specific threats for species and populations in each city through long-term monitoring schemes is the first step to ensure these conservation goals in the medium and long-term. The local governments can boost campaigns of citizen science to increase awareness and collect biodiversity data to improve local species checklists and field-based population studies of targeted taxa. In addition, further studies may assess how species' traits respond to urbanization and climate change at the local scales [81, 82]. These studies are necessary to evaluate how different taxa populations respond to the ongoing urban heat island effects.

Another element that needs to be assessed how urban protected areas are working across gradients of urbanization [83], climate change, and UHI. In particular, it is necessary to conduct studies to establish the effectiveness of these conservation areas inside cities and how they can improve in the face of future climate change and other human stressors. Local

governments, civil organizations, and academia in each municipality should work together to elaborate on conservation plans and define local actions to improve the effectiveness of these urban conservation areas (e.g., reducing urban heat island effects). For instance, these elements can help us identify the shape, size, and location of additional protected areas for targeted taxa and reinforce the effectiveness of urban green corridors for dispersal [13]. Such knowledge would be helpful in guiding strategies to decide how to expand urban areas, integrating explicitly nature's contributions to people (NCP) to alleviate the problem of urban heat islands. Accordingly, some specific activities should be implemented quickly to reduce biodiversity loss and heat stress. For instance, removing feral and stray dogs and cats should be a priority for local Mexican authorities in the next years due to the well-known impacts of these species on urban wildlife and people's health [84]. Further, implementing extensive tree plantation programs and creating extensive green spaces in low-income neighborhoods across Mexican municipalities (particularly those in higher risk; Fig 3) is key for adapting to the ongoing regional-local climate change impacts. These kinds of local actions are win-win strategies with multiple benefits simultaneously because they reduce heat stress on the human population and increase local urban biodiversity. Besides, we need to remember that urban areas represent only 1.0% of the terrestrial surface of our planet [85], but the indirect impacts go well beyond that share. In Mexico, urban areas account for ~1.2% [86], but the influence of cities on biodiversity is not limited to these crowded areas. Further studies are necessary to assess the effects of urban externalities on rural areas.

## Conclusions

Our results suggest that urban planning in Mexico should explicitly consider the need to implement mitigation and conservation actions. Mexican municipalities with less than five species were variable for each taxonomic group (Table C in S1 Data). Additional surveys in these municipalities will be necessary to establish whether this low number of species is not a sampling artifact. The rarefaction analysis gave us a comprehensive dataset of urban biodiversity for a large set of Mexican municipalities (389). We call for further studies with local surveys at finer scales to understand better the ecological and evolutionary processes shaping urban biodiversity in Mexico.

Some conservation actions might be implemented quickly to improve habitat diversity (e.g., restoration or conservation planning) and prioritize those municipalities with small urban areas and low species richness. As urbanization sprawls towards the periphery of towns, where remnant vegetation tends to exist, it will be necessary to consider those sites as potential protected areas and study the possibility of connecting them through green corridors of native vegetation. Such urban green spaces need restoration and expansion to mitigate UHI's impacts (and additional risks such as air quality, noise pollution, etc.), habitat fragmentation, and climate change. Besides, data on species and populations should increase and improve through long-term monitoring to evaluate the importance of protected areas and green spaces in cities for different species and higher taxa. Finally, Mexican urban planners should interact more with conservation biologists, social scientists, and climate change experts to help design and restructure cities in Mexico. Integrating disciplines would be crucial to alleviate biodiversity loss as urbanization and climate change processes occur.

## Supporting information

**S1 Appendix. Supplementary figures.**
(PDF)

**S1 Data. Supplementary tables.**
(XLSX)

## Acknowledgments

We thank Robbie Burger and three anonymous reviewers for their very helpful comments on previous drafts of this manuscript.

## Author Contributions

**Conceptualization:** Julián A. Velasco, Francisco Estrada, Constantino González-Salazar.

**Data curation:** Julián A. Velasco, Carlos Luna-Aranguré, Oscar Calderón-Bustamante, Alma Mendoza-Ponce, Constantino González-Salazar.

**Formal analysis:** Julián A. Velasco, Alma Mendoza-Ponce, Francisco Estrada, Constantino González-Salazar.

**Investigation:** Julián A. Velasco.

**Methodology:** Julián A. Velasco.

**Visualization:** Oscar Calderón-Bustamante, Alma Mendoza-Ponce.

**Writing – original draft:** Julián A. Velasco, Francisco Estrada.

**Writing – review & editing:** Julián A. Velasco, Carlos Luna-Aranguré, Alma Mendoza-Ponce, Francisco Estrada, Constantino González-Salazar.

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
