## [Decision Letter · Decision Letter 0]

18 Mar 2024

PONE-D-24-06618Drivers of urban biodiversity in Mexico and joint risks from future urban expansion, climate change, and urban heat island effectPLOS ONE

Dear Dr. Gonzalez-Salazar,

Thank you for submitting your manuscript to PLOS ONE. After careful consideration, we feel that it has merit but does not fully meet PLOS ONE’s publication criteria as it currently stands. Therefore, we invite you to submit a revised version of the manuscript that addresses the points raised during the review process.

We look forward to receiving your revised manuscript.

Kind regards,

Bijeesh Kozhikkodan Veettil

Academic Editor

PLOS ONE

Journal Requirements:

"Universidad Nacional Autónoma de México"

"We acknowledge financial support from the Programa de Investigación en Cambio Climático at UNAM (PINCC-UNAM) through a grant project to JAV. We thanks to Robbie Burger for very helpful comments on previous drafts of this manuscript."

"Universidad Nacional Autónoma de México"

6. We note that [Figures 2, S2, S3, S4 and S6] in your submission contain [map/satellite] images which may be copyrighted. All PLOS content is published under the Creative Commons Attribution License (CC BY 4.0), which means that the manuscript, images, and Supporting Information files will be freely available online, and any third party is permitted to access, download, copy, distribute, and use these materials in any way, even commercially, with proper attribution. For these reasons, we cannot publish previously copyrighted maps or satellite images created using proprietary data, such as Google software (Google Maps, Street View, and Earth). For more information, see our copyright guidelines: http://journals.plos.org/plosone/s/licenses-and-copyright.

a. You may seek permission from the original copyright holder of Figures 2, S2, S3, S4 and S6 to publish the content specifically under the CC BY 4.0 license.  

Reviewers' comments:

Reviewer's Responses to Questions

**Comments to the Author**

1. Is the manuscript technically sound, and do the data support the conclusions?

Reviewer #1: Yes

Reviewer #2: Partly

Reviewer #3: No

2. Has the statistical analysis been performed appropriately and rigorously? 

Reviewer #1: Yes

Reviewer #2: I Don't Know

Reviewer #3: No

3. Have the authors made all data underlying the findings in their manuscript fully available?

Reviewer #1: Yes

Reviewer #2: Yes

Reviewer #3: No

4. Is the manuscript presented in an intelligible fashion and written in standard English?

Reviewer #1: Yes

Reviewer #2: No

Reviewer #3: Yes

5. Review Comments to the Author

Reviewer #1: 1. The introduction part suggests to express the shortcomings of previous research and the innovation points of this research more clearly. In addition, a framework diagram of the article can be supplemented.

2. The article is too long, so the number of words can be appropriately compressed.

3. To discuss the Drivers of urban biodiversity section, it is recommended to add a graph to intuitively reflect the results.

4. The research shortcomings and prospects in the discussion and conclusion sections are not clear enough.

Reviewer #2: While I understand the inspiration and motivation behind this paper and believe that it is attempting to address important topics, I did not find it to be a particularly easy to follow narrative. There is very heavy reliance on ecological terminology without adequate contextualization or explanation that, I believe, will make it challenging to understand for a variety of audiences. I highly suggest defining specific terms for clarity throughout the text and generally addressing the approachability of the language. Narratively, this paper, as it stands, is challenging to follow…I did not find the goals of this study to be clearly outlined in the introduction, nor appropriately addressed in the conclusions. I highly recommend authors address both of these sections in particular. While I did find the presentation of the impacts of specific variables on biodiversity across municipalities very interesting, I did not see much tie back to concrete ecological relevance of these variables or specific and obtainable conservation actions to be taken, beyond the collection of more and better data. I believe one of the goals of this paper was to make specific conservation recommendations, so I would make those recommendations clearer. While I understand that making broad scale assumptions regarding the impact of variables on biodiversity across cities can be an important first step in making urban wildlife conservation recommendations, I would like to see more of an effort to address these municipalities less homogeneously. Again, I think providing greater ecological context to the municipalities analyzed and/or the clusters into which they were placed and discussing more clearly the variability across municipalities/clusters would greatly bolster this paper. if there were specific limitations to this study (i.e. data availability) that prevent looking at variability between municipalities in greater detail, I would recommend making those limitations clearer. Additionally, as you'll see in my attachment, I wish it were clearer how the authors defined urban vs. rural areas and why they decided to rely on a binary of urban vs. rural instead of using more of a gradient of urbanization based on clearly defined variables, as many other urban biodiversity studies have done. Again, if creating an urbanization gradient was not possible for this study, I recommend explaining why.

Reviewer #3: The topic of the paper is quite timely and important.

I really appreciate the effort and time you've invested in your research.

While this study addresses an important topic, I have concerns regarding the methodology and the variables used. In my opinion, especially the socioeconomic variables employed may not be directly aligned with answering the research question posed in your study. Research needs to utilize variables that directly contribute to addressing the research question and achieving the study's objectives. It's tough to model GDP. Also, imagine a new city with a focus on conservation might have lower tree heights (being a newly planned city), giving a false positive. Similarly, a factory town might have a higher GDP but lower bio-diversity. There is enough literature on the characteristics of factory towns.

In this case, the variables employed may not sufficiently capture the intended phenomenon or provide the necessary insights to draw meaningful conclusions.

I agree with the use of the Random Forest Model, but the use of PCA is not explained fully. The LUCC model is a good choice and should be explored further.

Additionally, I recommend conducting a thorough literature review to identify relevant variables validated and utilized in similar studies.

6. PLOS authors have the option to publish the peer review history of their article (what does this mean?). If published, this will include your full peer review and any attached files.

Reviewer #1: No

Reviewer #2: No

Reviewer #3: No

---

## [Author Response · Author response to Decision Letter 0]

9 May 2024

TO EDITOR

We thank the reviewers for their significant efforts in reviewing and suggesting improvements to our paper. We tried to attend their suggestions and concerns. In this new version, we improved the manuscript, including aspects suggested by the reviewers. We deeply appreciate their constructive criticisms. In this document, we have provided responses to each of the comments.

 Comments from the Editor

1- Please ensure that your manuscript meets PLOS ONE's style requirements, including those for file naming.

R/: The document and figures have been updated.

Datasets and R codes can be accessed at this link: https://figshare.com/s/f7e1d3d9d9883c63e23d (private link for external review; it will be updated upon being accepted).

3- We note that the grant information you provided in the ‘Funding Information’ and ‘Financial Disclosure’ sections do not match. When you resubmit, please ensure that you provide the correct grant numbers for the awards you received for your study in the ‘Funding Information’ section.

R/: We will add the The Programa de Investigación en Cambio Climático at UNAM (PINCC-UNAM) in the Funding Information section, but no award numbers were supplied for these grants. We will update the Award Number section to read “n/a” in the submission portal.

4- Thank you for stating the following financial disclosure: "Universidad Nacional Autónoma de México" 

R/: The funders had no role in study design, data collection and analysis, decision to publish, or preparation of the manuscript.

"Universidad Nacional Autónoma de México"

R/: We have removed the Programa de Investigación en Cambio Climático at UNAM (PINCC-UNAM) acknowledgment from the manuscript.

Funding Statement should be updated as:

1. Universidad Nacional Autónoma de México 

2. The Programa de Investigación en Cambio Climático at UNAM (PINCC-UNAM), grant number “n/a”

6- We note that [Figures 2, S2, S3, S4 and S6] in your submission contain [map/satellite] images which may be copyrighted. All PLOS content is published under the Creative Commons Attribution License (CC BY 4.0), which means that the manuscript, images, and Supporting Information files will be freely available online, and any third party is permitted to access, download, copy, distribute, and use these materials in any way, even commercially, with proper attribution.

R/: We have revised the figures so that all information shown is licensable under CC-BY 4.0. The maps are created with data that is permissible for publication. Data sources are now listed in the figure captions. In addition, the figure numbering has changed because of the paper restructuring and removal of figures 1 and 2. The mapping of figure numbers between revisions is shown below:

Fig 3 is now Fig 1, Fig 4 is now Fig 2, Fig 5 is now Fig 3, and Fig 6 is now Fig 4.

REPLY TO THE REVIEWERS

Reviewer #1: 

1. The introduction part suggests to express the shortcomings of previous research and the innovation points of this research more clearly. In addition, a framework diagram of the article can be supplemented.

R/: Thanks for this comment. We have added a final paragraph describing the study's objectives in the introduction and made some changes to highlight the relevance of our research, which is one of the first to establish how urban biodiversity varies across an extensive set of municipalities in the same country (Lines 132-134).

2. The article is too long, so the number of words can be appropriately compressed.

R/: We agree with the recommendation. We have made the necessary changes to reduce the article length. We have attempted to be more concise, and we have also removed several elements. For instance, we removed the PCA and cluster analyses following the recommendations from the other reviewers to improve clarity in our analysis and questions that we want to address here. We expect that this new version will be more precise and fluid for readers.

3. To discuss the Drivers of urban biodiversity section, it is recommended to add a graph to intuitively reflect the results.

R/: Thanks for this comment. We think that Fig 2 of effect size may be a visual guidance of the discussion on the variables evaluated here as we are testing the direct effect of each predictor on urban species richness.

4. The research shortcomings and prospects in the discussion and conclusion sections are not clear enough.

R/: Thanks for this comment. We have improved the discussion and conclusion sections to clarify the limitations found in our study, as well as to present research perspectives that may arise from this type of study.

Reviewer #2: 

While I understand the inspiration and motivation behind this paper and believe that it is attempting to address important topics, I did not find it to be a particularly easy to follow narrative. There is very heavy reliance on ecological terminology without adequate contextualization or explanation that, I believe, will make it challenging to understand for a variety of audiences. I highly suggest defining specific terms for clarity throughout the text and generally addressing the approachability of the language. Narratively, this paper, as it stands, is challenging to follow…I did not find the goals of this study to be clearly outlined in the introduction, nor appropriately addressed in the conclusions. I highly recommend authors address both of these sections in particular. 

R/: Thanks for these comments. We have attempted to rewrite several sections of the manuscript to provide the context required for a better understanding of our study. In the introduction, we have added a final paragraph describing the study's objectives and made some changes to highlight the relevance of our research. We expect that this new version will be more precise and fluid for readers.

While I did find the presentation of the impacts of specific variables on biodiversity across municipalities very interesting, I did not see much tie back to concrete ecological relevance of these variables or specific and obtainable conservation actions to be taken, beyond the collection of more and better data. I believe one of the goals of this paper was to make specific conservation recommendations, so I would make those recommendations clearer. 

R/: Thanks for these comments. According to our results, we have made some specific conservation recommendations in the discussion and conclusions. Specifically, we mentioned the need to implement mitigation actions to reduce local climate change effects through the habitat recovery or delimitation of new urban protected areas (lines 435-457). These recommendations are derived from our analysis, which shows that municipalities with small urban areas and low species richness should be considered a conservation priority (lines 444-446). 

While I understand that making broad scale assumptions regarding the impact of variables on biodiversity across cities can be an important first step in making urban wildlife conservation recommendations, I would like to see more of an effort to address these municipalities less homogeneously. Again, I think providing greater ecological context to the municipalities analyzed and/or the clusters into which they were placed and discussing more clearly the variability across municipalities/clusters would greatly bolster this paper. if there were specific limitations to this study (i.e. data availability) that prevent looking at variability between municipalities in greater detail, I would recommend making those limitations clearer. 

R/: Thanks for these comments. We mentioned in the results that municipalities exhibiting a high multivariate risk from future urban sprawl, the urban heat island, and climate change are concentrated in northern Mexico (lines 306-330). We have now included that mitigation and habitat-restoring actions should focus on these municipalities specifically (lines 410-412).

Additionally, as you'll see in my attachment, I wish it were clearer how the authors defined urban vs. rural areas and why they decided to rely on a binary of urban vs. rural instead of using more of a gradient of urbanization based on clearly defined variables, as many other urban biodiversity studies have done. Again, if creating an urbanization gradient was not possible for this study, I recommend explaining why.

R/: Thanks for these comments. In the material and methods section, we included how urban vs. rural areas were defined in our study (Lines 123-139). Unfortunately, we do not have data to create an urbanization gradient across Mexican municipalities. In addition, most socio-economic data used by us come from this urban delimitation conducted by the Mexican government (Lines 138-139). 

Reviewer #3: 

The topic of the paper is quite timely and important.

I really appreciate the effort and time you've invested in your research.

R/: Thanks for these comments.

While this study addresses an important topic, I have concerns regarding the methodology and the variables used. In my opinion, especially the socioeconomic variables employed may not be directly aligned with answering the research question posed in your study. Research needs to utilize variables that directly contribute to addressing the research question and achieving the study's objectives. It's tough to model GDP. Also, imagine a new city with a focus on conservation might have lower tree heights (being a newly planned city), giving a false positive. Similarly, a factory town might have a higher GDP but lower bio-diversity. There is enough literature on the characteristics of factory towns.

R/: Thanks for these comments. We used several socioeconomic variables identified in the literature as the most relevant to explain urban species richness. In table 1, we detailed predictions of each variable and an explanation of how a given variable is linked with species richness. Although GDP can be hard to quantify, we used socio-economic datasets from the Mexican government, so we consider that this variable has been measured with the same bias across all Mexican municipalities. In addition, we did not find evidence that GDP explains urban species richness across Mexican cities.

In this case, the variables employed may not sufficiently capture the intended phenomenon or provide the necessary insights to draw meaningful conclusions.

R/: Thanks for these comments. However, we have recognized in the discussion that more fine-scale variables must be included to explore in more detail how socioeconomic factors drive urban biodiversity in Mexican cities (lines 578-586). To our knowledge, this is the first study conducted in Mexico to explore how different variables may explain observed species richness for terrestrial vertebrates. Further studies on small spatial scales should be undertaken to establish the scale dependence of these variables.

I agree with the use of the Random Forest Model, but the use of PCA is not explained fully. The LUCC model is a good choice and should be explored further.

R/: We appreciate these comments and decided to exclude PCA and cluster analysis from our manuscript due to the lack of justification in the previous version. We consider that PCA and cluster analyses do not fit to the questions that we want to address for this study. 

Additionally, I recommend conducting a thorough literature review to identify relevant variables validated and utilized in similar studies.

R/: Thanks for pointing out this. We fully explained how the different variables we used have been used previously in similar studies (see Table 1). In particular, table 1 provides detailed information about these variables with appropriate references.

---

## [Decision Letter · Decision Letter 1]

10 Jun 2024

PONE-D-24-06618R1Drivers of urban biodiversity in Mexico and joint risks from future urban expansion, climate change, and urban heat island effectPLOS ONE

Dear Dr. Gonzalez-Salazar,

Thank you for submitting your manuscript to PLOS ONE. After careful consideration, we feel that it has merit but does not fully meet PLOS ONE’s publication criteria as it currently stands. Therefore, we invite you to submit a revised version of the manuscript that addresses the points raised during the review process.

We look forward to receiving your revised manuscript.

Kind regards,

Bijeesh Kozhikkodan Veettil

Academic Editor

PLOS ONE

Journal Requirements:

Reviewers' comments:

Reviewer's Responses to Questions

**Comments to the Author**

1. If the authors have adequately addressed your comments raised in a previous round of review and you feel that this manuscript is now acceptable for publication, you may indicate that here to bypass the “Comments to the Author” section, enter your conflict of interest statement in the “Confidential to Editor” section, and submit your "Accept" recommendation.

Reviewer #4: All comments have been addressed

2. Is the manuscript technically sound, and do the data support the conclusions?

Reviewer #4: Yes

3. Has the statistical analysis been performed appropriately and rigorously? 

Reviewer #4: Yes

4. Have the authors made all data underlying the findings in their manuscript fully available?

Reviewer #4: Yes

5. Is the manuscript presented in an intelligible fashion and written in standard English?

Reviewer #4: Yes

6. Review Comments to the Author

**Reviewer #4: **Overall, this is a clear, concise, and well-written manuscript. The introduction is relevant and theory-based, providing sufficient information about previous study findings for readers to follow the present study rationale and procedures. The study contributes valuable insights into Drivers of urban biodiversity in Mexico and joint risks from future urban expansion, climate change, and urban heat island effect.

Detailed Comments

Title and Abstract

• The title accurately reflects the content of the manuscript.

• The abstract is clear and effectively summarizes the study’s main points.

Introduction

• The introduction is relevant and grounded in theory.

• It provides a clear rationale for the study, supported by previous findings.

Literature Review

• The literature review is comprehensive and includes relevant studies.

• It effectively situates the current research within the existing body of work.

Methods

• The methods are well-detailed and appropriate for the study’s objectives.

• They are described clearly, allowing for reproducibility.

Results

• The results are presented clearly and logically.

• Tables and figures are used effectively to illustrate key points.

Discussion and Conclusion

• The discussion is thorough and interprets the findings in the context of existing research.

• The conclusions are supported by the data presented.

References

• References are accurate and relevant to the study.

Specific Comments

• The manuscript is well-written, with only minor grammatical errors noted.

• Figures and tables are of high quality and appropriately labeled.

Recommendation

I recommend minor revisions. The manuscript is strong overall but would benefit from addressing the following points:

• Detail the Methodologies: Provide more detailed information on the methodologies used to assess urban biodiversity, climate change impacts, and the urban heat island effect. This includes specifying data sources, data collection methods, and analytical techniques. Clear and comprehensive methodological details will enhance the reproducibility and reliability of your findings.

• Expand the Literature Review: Broaden the literature review to include more recent studies on urban biodiversity, particularly those relevant to Mexico. This will help to better contextualize your research within the existing body of knowledge and highlight the unique contributions of your study. Including local studies will also strengthen the relevance and applicability of your findings to Mexico's specific context.

• Enhance the Discussion of Implications: Expand the discussion section to provide a deeper interpretation of your findings and their broader implications for urban planning and biodiversity conservation in Mexico. Specifically, discuss how the results can inform policy decisions, urban development strategies, and conservation efforts to mitigate the joint risks from urban expansion, climate change, and the urban heat island effect. This will help to highlight the practical significance of your study and its potential impact on urban policy and environmental management.

7. PLOS authors have the option to publish the peer review history of their article (what does this mean?). If published, this will include your full peer review and any attached files.

Reviewer #4: **Yes: **Bhumika Das

---

## [Author Response · Author response to Decision Letter 1]

23 Jul 2024

TO EDITOR

We thank the reviewer for their significant efforts in reviewing and suggesting improvements to our paper. We appreciate very much their comments. Furthermore, we tried to attend their suggestions and concerns. In this document, we have provided responses to each of their comments.

REPLY TO REVIEWER

1. Detail the Methodologies: Provide more detailed information on the methodologies used to assess urban biodiversity, climate change impacts, and the urban heat island effect. This includes specifying data sources, data collection methods, and analytical techniques. Clear and comprehensive methodological details will enhance the reproducibility and reliability of your findings.

R/: Thank you for this comment. We have attempted to rewrite the methods section to provide the context needed for a better understanding of our study. Additionally, we made data and R codes available at this link: https://figshare.com/s/f7e1d3d9d9883c63e23d to ensure the reproducibility of our analysis.

2. Expand the Literature Review: Broaden the literature review to include more recent studies on urban biodiversity, particularly those relevant to Mexico. This will help to better contextualize your research within the existing body of knowledge and highlight the unique contributions of your study. Including local studies will also strengthen the relevance and applicability of your findings to Mexico's specific context.

R/: Thanks for their suggestion. We have incorporated additional references from recent studies on Mexican biodiversity to contextualize our work and emphasize its significance.

3. Enhance the Discussion of Implications: Expand the discussion section to provide a deeper interpretation of your findings and their broader implications for urban planning and biodiversity conservation in Mexico. Specifically, discuss how the results can inform policy decisions, urban development strategies, and conservation efforts to mitigate the joint risks from urban expansion, climate change, and the urban heat island effect. This will help to highlight the practical significance of your study and its potential impact on urban policy and environmental management.

R/: Thanks for this comment. We have made the necessary changes, improving the discussion section to present research perspectives that may arise from this type of study, as well as to support conservation strategies to avoid urban biodiversity loss and improve the sustainable agendas of Mexican cities.

---

## [Editor Report · Decision Letter 2]

25 Jul 2024

Drivers of urban biodiversity in Mexico and joint risks from future urban expansion, climate change, and urban heat island effect

PONE-D-24-06618R2

Dear Dr. Gonzalez-Salazar,

We’re pleased to inform you that your manuscript has been judged scientifically suitable for publication and will be formally accepted for publication once it meets all outstanding technical requirements.

Kind regards,

Bijeesh Kozhikkodan Veettil

Academic Editor

PLOS ONE
---

## [Editor Report · Acceptance letter]

12 Aug 2024

PONE-D-24-06618R2 

PLOS ONE

Dear Dr. Gonzalez-Salazar, 

I'm pleased to inform you that your manuscript has been deemed suitable for publication in PLOS ONE. Congratulations! Your manuscript is now being handed over to our production team.

Kind regards, 

on behalf of

Dr. Bijeesh Kozhikkodan Veettil 

Academic Editor

PLOS ONE